# Puzzle of Proteoform Variety—Where Is a Key?

**DOI:** 10.3390/proteomes12020015

**Published:** 2024-05-10

**Authors:** Stanislav Naryzhny

**Affiliations:** B. P. Konstantinov Petersburg Nuclear Physics Institute, National Research Center “Kurchatov Institute”, Leningrad Region, Gatchina 188300, Russia; snaryzhny@mail.ru

**Keywords:** two-dimensional gel electrophoresis, proteoforms, mass-spectrometry, degradome

## Abstract

One of the human proteome puzzles is an imbalance between the theoretically calculated and experimentally measured amounts of proteoforms. Considering the possibility of combinations of different post-translational modifications (PTMs), the quantity of possible proteoforms is huge. An estimation gives more than a million different proteoforms in each cell type. But, it seems that there is strict control over the production and maintenance of PTMs. Although the potential complexity of proteoforms due to PTMs is tremendous, available information indicates that only a small part of it is being implemented. As a result, a protein could have many proteoforms according to the number of modification sites, but because of different systems of personal regulation, the profile of PTMs for a given protein in each organism is slightly different.

## 1. Introduction

One of the most significant recent breakthroughs in proteomics is the discovery of an unexpected level of complexity in the human proteome. Interestingly, on one side, the number of protein-coding genes turned out to be much smaller than expected. Initially, the number of proposed human protein genes was more than 100,000 [1]. Now, it is estimated to be only 19,778 (April 2023) coded in the nuclei by 46 chromosomes and 13 coded by mitochondrial DNA. But on the other side, different protein molecules can be produced from a single gene [2]. The components of this variety are called proteoforms and compose the whole human proteome. The main aim of researchers involved in the Human Proteome Project (HPP) organized by the Human Proteome Organization (HUPO) is “to map the entire human proteome” (https://hupo.org/mission) (accessed on 22 December 2023). And the grand challenge of the project is to decipher “a function for every protein” (https://hupo.org/TheGrandChallenge) (accessed on 22 December 2023). During the 20 years since the start of the HPP, especially in the last 10 years, the more complicated vision of the human proteome was formed. When we talk about a protein’s function, we should keep in mind the complexity of each protein. The name “protein” is actually an umbrella covering sometimes functionally different molecules called proteoforms [3]. 

## 2. Standardization Aspects

For some period, many words were used (and still some can be found) for the diversity of protein molecules: “protein forms”, “protein isoforms”, “protein species”, “protein variants”, and “mod forms”. The term “isoform” or “protein variant” is possibly the most popular one. But sometimes it can have a slightly different meaning. The oldest dictionary publisher in the United States, Merriam-Webster, gives a definition of isoform based on the sequence: “any of two or more functionally similar proteins that have a similar but not identical amino acid sequence” https://www.merriam-webster.com/ (accessed on 22 December 2023). There is another definition as follows: “an isoform is a member of a set of highly similar proteins that originate from a single gene or gene family and are the result of genetic differences” [4]. If we consider not only a single gene but a gene family as a source of different isoforms, it will generate some kind of uncertainty. Such a meaning is used mainly in enzymology, where the term “isoform” is used in line with the term “enzyme isoform” or “isozyme” [5]. It is a bit confusing, and a gene-centric definition is more appropriate. The UniProt Knowledgebase defines an isoform as “a protein form that is generated due to alternative splicing, variable promoter usage, or other post-transcriptional variations of a single gene” [6]. Accordingly, IUPAC (International Union of Pure and Applied Chemistry) defines isoforms only based on genetic differences [4]. 

The term “protein species” was initially introduced in 1996 by Peter Jungblut to explain many spots of the same protein after separation by two-dimensional gel electrophoresis (2DE) [7,8]. For instance, fifty-nine spots were stained with Hsp27 (HSPB1) antibodies on a high-resolution 2DE blot [9]. This term was used in proteomics for a long time until Neil Kelleher proposed a new one: “proteoform” [10]. It has practically the same meaning as “protein species” and is used to designate “all the different molecular forms in which the protein product of a single gene can be found, including changes due to genetic variations, alternatively spliced RNA transcripts, and PTMs” [11]. The classical scheme for the generation of proteoforms is shown in Figure 1 [12].

Gradually, the term “proteoform” became more popular than “protein species” and became a commonly accepted term to be used in publications about protein variety. Figuratively speaking, according to the authors, it is “proteomics currency” now [13]. This is just an example of how one term gains popularity and becomes standard, but another does not. But the aspect of terminology is only one side of the situation. Another side is the mainstream study of proteoforms. Two approaches based on mass spectrometry exist: top-down and bottom-up. By using the top-down approach, the native molecular mass of a proteoform is directly measured by MS, allowing it to definitely identify the PTM status of the proteoform [14,15]. 

The Consortium for Top-Down Proteomics initiated the “Human Proteoform Project” in 2021. The aim is grandiose to interpret the full range of diverse proteoforms generated from all genes in the human genome [16]. The consortium developed rules for writing a definite proteoform. As they say, “this nomenclature is intended to be both machine- and human-readable and to be sufficiently flexible to meet current and foreseeable needs”. For recording the sequence of fully characterized proteoforms, they use a standardized notation, “ProForma”, that “provides a means to convey any proteoform by recording the amino acid sequence using standard single-letter notations and indicating modifications or unidentified mass shifts in parentheses after certain amino acids” [17,18]. Accordingly, data on various proteoforms obtained by top-down proteomics are being included in the Proteoform Atlas [19]. 

The bottom-up MS data are redundant and do not suit proteoform identification. To figure out possible proteoforms, the bottom-up MS data need to be additionally treated [20]. But still, to exclude ambiguity, the preliminary selection (separation) of specific proteoforms is needed. Classically, it can be performed by 2DE. What is more, based on 2DE, the molecular weight (Mw) and pI of the proteoforms can be measured. After specific hydrolysis, further analyses can be performed by the bottom-up MS [21,22]. Actually, based on 2DE separation followed by bottom-up MS, the proteoform profiles were generated for several types of cells [23,24,25,26]. What is more, these data were used to generate a web database called “2DE-pattern” [27]. The data representation here is based on the felicitous visual properties of 2DE gels. An example of such a protein inventory is shown in Figure 2. Though this approach is not as exact as the top-down MS, it gives a general visual representation of the families of proteoforms (2DE patterns). Despite these attractive qualities, 2DE still remains a kind of art that requires a lot of effort and time to perform [28,29]. Because of this, there are relatively few labs in the world that are using 2DE. Accordingly, it can be a hurdle in the usage of data presented in 2DE databases for labs not dealing with 2DE (https://world-2dpage.expasy.org/portal/) (accessed on 22 December 2023). To overcome this, these 2DE databases should be connected to other databases that are more popular, such as Swiss Prot/Uniprot, Nextprot, Human Protein Atlas, etc. [30,31,32]. 

Both approaches (top-down MS and bottom-up MS in combination with 2DE, or “integrative top-down proteomics”) have advantages and disadvantages. What is more important is that they are complimentary to each other [21]. The main problem now is how to unify the results of these investigations. It could be an overall benefit if a solution for data standardization and unification is found. At least some information can be added to the Proteoform Atlas to make the data possible for comparison between databases. For instance, an average Mw and the calculated isoelectric point (pI) of each proteoform deposed into the Proteoform Atlas could make a better connection between the data in the Proteoform Atlas and the database “2DE-pattern”. Another significant aspect is the size of the polypeptides. There are very small polypeptides (some are even below 50 AA) deposed into the Proteoform Atlas as proteoforms. Many of them are functional products of proteolytic processing. For instance, the removal of the N-terminal methionine and the signal peptide is essential for the correct maturation and secretion of many proteins. Through cleavage of domains and processing, inactive proteins can be converted into active forms, or vice versa [33]. An interesting example is represented by pro-opiomelanocortin (POMC). Here, the removal of the 26-AA signal peptide produces the 241-AA polypeptide, which undergoes a series of PTMs such as phosphorylation and glycosylation, before being proteolytically cleaved by endopeptidases into 11 chains with different physiological activities [33]. Many proteoforms deposited in the Proteoform Atlas are small polypeptides that are not generated by processing but are likely a result of degradation by proteasomes (for instance, fragments of actin). An example of such a situation is presented in Figure 3 [34]. 

Proteasomes are the barrel-like complexes that degrade proteins and deprive them of functionality. These complexes possess caspase-like (β1), trypsin-like (β2), and chymotrypsin-like (β5) proteolytic activities and degrade proteins through ubiquitin-dependent or -independent pathways [35,36]. As a result of the protein turnover, a so-called “degradome” is generated [37,38,39]. Actually, this terminology can be a bit confusing, as the term “degradome” is also used for the definition of the whole set of cellular proteases [40]. In our case, the degradome is a part of the peptidome that is defined as a population of low-molecular-weight biologic peptides. These peptides are critical for normal cellular and organismal functions. In addition, the peptidome also contains fragments of larger proteins produced by normal or abnormal degradation (the degradome) [41]. It is important that this subset of the peptidome is an attractive target in cancer research, for instance, as a biomarker of cancer metastasis [37]. Altogether, proteasomal degradation is an extra type of PTM regulation that controls biological activity and the fate of cells [42,43]. 

If we are going to consider these degradation products as proteoforms, the number of possible proteoforms will dramatically increase [44]. It seems this issue needs to be discussed, as a lot of these products are listed in the Proteoform Atlas. If we accept it, the general scheme of proteoform generation and turnover will look like Figure 4. 

## 3. Quantification of Proteoforms

So far, the main task of the Human Proteome Project has been a mapping of the human proteome by finding and describing all proteins. Now, the more challenging task is to decipher the whole proteome’s complexity, including thousands of proteoforms [21]. This is much more complicated work, as we do not yet know the number of proteoforms in a human proteome [2,45]. We can only make some extrapolations and calculations based on the available data. But, exact numbers can be very different depending on the applied approximation. The main problem is instrumentation sensitivity, which should allow for measuring molecules at a concentration of a single copy per cell [46]. The sensitivity of mass spectrometric analysis is a key factor here. There is significant progress in this area, but the requirements of reality still greatly exceed the capabilities of modern mass spectrometers. Over the past decade, new mass spectrometers have been developed with increased sensitivity, reliability, and specificity. New methods are needed to further improve MS performance for accurate qualitative and quantitative analyses. At the same time, improved pre-treatment technology and ionization technology can be combined. The rapid development of MS will promote its application in various fields such as clinical trials, environmental monitoring, life sciences, etc. [47,48,49]. A new ultra-high sensitivity LC–MS workflow has enabled proteome analysis of single cells [48]. Studies at the proteoform level need higher sensitivity.

This is only part of the issue. Another part is the proteoform pattern or profile complexity itself, which can be very different for different proteins. In other words, different proteins can exist in different numbers of proteoforms. What is interesting is that the range of proteoforms per protein is very wide. The most reliable method for proteoform separation and detection is 2DE in combination with ESI LC–MS/MS. For instance, Thiede et al. identified and quantified 1245 proteins from 2711 spots in HeLa cells [50]. It was shown that only ~50% of the proteins (431) were found in one 2DE spot each, and 174 proteins were found in only two spots each. They also found 16 proteins in multiple 2DE spots (≥20) (Figure 5). Actin was at the top—54 spots. Similar results were obtained from glioblastoma cells and HepG2 cells using the classical spot-picking approach as well as the sectional 2DE with the following ESI LC–MS/MS [11]. In all cases, we see a similar distribution of proteoform numbers between different proteins. The main portion of proteins has only one–two proteoforms, but others can have much more—up to a hundred (Figure 5).

There is another aspect that also needs to be mentioned here. Some spots contain more than just one protein. It shows that there is a redundancy of parameters (pI/Mw). Despite the high resolution of 2DE, some proteoforms originated from different genes can be located in the same position because of very similar parameters (pI/Mw) [22,51,52]. Though this situation can be easily improved using very narrow pH gradients, a resolution of up to 0.001 pH units can be achieved [51]. 

The main input in proteoform varieties is PTM. About 5% of the proteome comprises enzymes that perform more than 400 types of PTMs [53] (http://www.unimod.org) (accessed on 22 December 2023). What is interesting is that PTMs in different proteins are not present uniformly. The number of PTM sites on a single protein can range from 0 to over 100. Here, 75% of proteins contain two or fewer PTMs, and only a few have more than one hundred [2]. What is more, the graphical distribution looks very similar to the graphs presented in Figure 5 (Figure 6). This confirms again that the main input in proteoform variety is performed by PTMs.

**Figure 5 proteomes-12-00015-f005:**
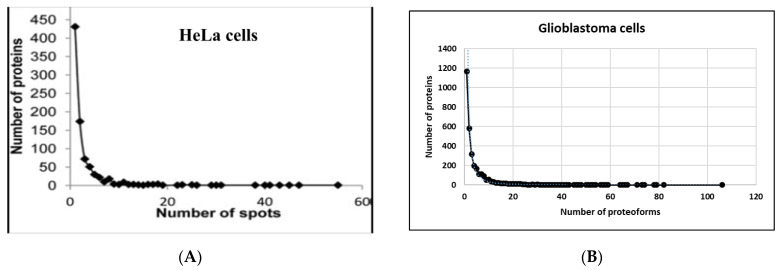
(**A**) The number of spots where a unique protein was detected. Adapted with permission from [50]. (**B**) The number of proteoforms that different proteins have. The figure was generated using data from [54].

## 4. Aspects of Generation of PTMs

There is another question: even considering strict regulation of proteoform numbers, why can some proteins, like actin or histones, be observed in hundreds of proteoforms, but others just in one or two? Why is there such discrimination between proteins? What is the reason for some of them to be so heavily modified? As the situation is very different for different proteins, the answer could be in the specific functionality of the proteins. If we disclosed the mode of functioning and origin of proteoforms, we could explain the reason for their diversity. We can go here step by step, considering the personality of each multi-proteoformic protein. For histones, PTMs work as a histone code that at least partially explains multiple modifications of histones [55]. For other proteins, we need to find another explanation for the presence of a high number of proteoforms. There is a chance to find a solution if we perform a bioinformatic analysis of these proteins (some of these proteins are presented in Table 1). For instance, analysis by Panther 18.0 (https://www.pantherdb.org/) (accessed on 22 December 2023) shows that according to the protein class, at the top of the list are proteins of the chaperone, cytoskeleton, and metabolism classes. But one reason for detecting more proteoforms for these proteins can be a sensitivity aspect. These proteins are mostly very abundant, for instance, such cytoskeleton proteins as actin or tubulins. Accordingly, there is a better chance to detect more forms of them. Heat shock proteins HS90A, HS90B, HSP7C, and ENPL belonging to the chaperone class are also among the most abundant cellular proteins. So, it seems that at least one reason for the detection of many proteoforms for some proteins is just their abundance. In favor of this view are the graphs of proteoform abundance inside the cell that follow Zipf’s law [45,56]. But this rule does not work for all proteins. For instance, tubulin alpha-8 chain (Q9NY65 · TBA8_HUMAN) or heat shock protein beta-8 (Q9UJY1 · HSPB8_HUMAN) have been detected so far only in one–two proteoforms. What is interesting is that according to Uniprot, both of these proteins have many possible PTM sites, 33 and 20, respectively. That means more proteoforms have not been detected so far just because of the sensitivity issue. Altogether, it seems that there is no direct connection between protein function and the number of proteoforms. 

It should be borne in mind that PTMs can appear at different points of the protein’s life cycle and have a range of half-lives. For example, many proteins are modified immediately after translation has completed, which ensures their correct structure or stability or directs the protein to distinct cellular compartments (e.g., nucleus, membrane). Other modifications occur at the sites of protein localization to influence its biological activity. The mechanism of binding to special tags ensures degradation, proteolytic processing, and a step-by-step mechanism for protein maturation or activation. PTMs can also be reversible, depending on the nature of the modification. For example, protein phosphokinases phosphorylate specific amino acid residues, a common method of catalytic activation or inactivation. On the contrary, phosphatases remove the phosphate group, again changing the biological activity of the protein.

Also, the personal landscape of PTMs can be dependent on health, age, environment, and other factors. For example, epigenetic regulation at the level of histone PTMs plays a major role in the aging process and affects lifespan. Pharmaceutical approaches to treat diseases associated with aging appear to be possible here [57]. The specific information on these PTMs can be found in the Uniprot database but are addressed in more detail in specialized databases such as the Aging Atlas, the Proteoform Atlas, or the Comparative Toxicogenomics Database [58,59,60].

## 5. Aspects of Proteoform Variety

The main proteomics puzzle is a discrepancy between the calculated and experimentally measured numbers of proteoforms. Considering the possibility of different PTMs at the same site, the number of theoretical proteoforms is huge. The possible number of different proteoforms in a cell of the same type is estimated at least 1,000,000 [2]. But, it seems that the theoretical combinatorial number for all possible variants is much bigger. For instance, the polypeptide that can be modified at 10 sites, according to Formula (1), could be present in more than 1000 different proteoforms.
(1)$$C_{n}^{k}=\frac{n!}{k!\times \left(n-k\right)!}$$

*n* is the number of all PTMs (10), and *k* is the number of combinations of PTMs (from 0 to 10).

Many proteins can have much more PTMs than 10 (Table 1). For instance, actin beta (P60709 · ACTB_HUMAN) can be phosphorylated, acetylated, ubiquitylated, etc., at more than 60 sites. Histone H3.2 (Q71DI3 · H32_HUMAN) can be phosphorylated, acetylated, ubiquitylated, etc., at more than 20 sites. Cellular tumor antigen p53 (P04637 · P53_HUMAN) can be modified at over 70 sites (https://www.phosphosite.org/) (accessed on 22 December 2023). The combinatorial calculation generates dozens of thousands of proteoforms for these proteins. Actually, by 2DE and mass spectrometry, p53 was detected in no more than 20 forms [61,62,63]. Using top-down mass spectrometry, it was shown that there were ~1000 differentially modified forms of actin beta and ~3000 histone H3.2 [64,65]. 

Even considering a sensitivity issue in detecting proteoforms, there is a big gap between the amounts of experimentally detected and theoretically possible proteoforms. It seems that there is a high degree of control over the enzymatic production and maintenance of PTMs [2]. Although the potential complexity of proteins due to PTMs is enormous, the available data suggest that only a small part of it is realized in each sample. But how this part is realized is another question. The main contradiction that is revealed when calculating the number of possible and actual detectable human proteoforms is most likely associated with individual variability. Of the total possible number of options, only a very limited part of them is implemented in each individual case. If we accept that all possible PTMs are realized in different people in a slightly different way, we can easily find all theoretical proteoforms. Here, there is a situation where each protein can have many proteoforms according to sites of modification, but because of different personal regulations, the patterns of PTMs that are realized in each person are different. Moreover, the implementation occurs in such a way that the main (major) proteoforms are produced in all individuals, but sets of many minor forms arise differently in everyone. Our recent study about the variety of proteoforms of the haptoglobin beta-chain is in favor of this hypothesis [66]. In this study, 2DE of proteins from the plasma of 20 donors, followed by immunological detection, revealed, in summary, 50 different proteoforms of the haptoglobin beta-chain. But in each sample, it was detected in no more than twenty forms, and only eight of the same forms (major) were present in each sample. So, we can assume that only the major forms are functional. Other (minor) forms and PTMs can be a product of stochastic noise and do not have a special effect on the functionality of the protein molecule. On the other hand, such a wide variety of proteoforms can serve as some kind of evolutionary mechanism. Despite all these assumptions, they require additional confirmation. As a minimum, the presence of such a strong variance at the level of plasma proteoforms can serve as an analogue of fingerprints at the molecular level and have practical significance.

## 6. Role of Bioinformatics

The main challenge of bioinformatics is to build a true understanding of processes from proteomics data [67,68,69]. Currently, many tools and databases are available for this purpose. For example, the Kyoto Encyclopedia of Genes and Genomes, BioCarta, GenMAPP, and PANTHER contain extensive information on metabolism, signal transduction, and interactions [70,71,72]. In addition, there are oncology-specific databases, such as Netpath [73]. Data about protein interactions can be found in BioGRID, IntAct, MINT, HRPD, or STRING [69,74,75,76]. Moreover, based on a list of the given proteins, these programs allow for drawing protein interaction networks [76]. 

But the point is, as we are going to decipher the details of protein functionality at the proteoform level, we need to transform all these platforms according to these needs. Knowing the gene name of the protein is not enough. As a minimum, the data about protein variety generated genetically at the isoform level should be included in the above-mentioned databases. The proteoforms are on the line.

## 7. Clinical Aspects of Proteoforms

It is necessary not only to take inventory of proteoforms but also to find out how they function, how proteoforms differ in different cell types, and how they change in diseases [77]. The assessment of PTMs itself is a very complex technical task. But the development of new and improved proteomics technologies makes it possible to solve it. Moreover, this is necessary in order to understand the functions that underlie many etiological processes [78,79]. This is where precision or personalized medicine can help better understand the many things that affect a patient’s health. Precision medicine is an approach to treating and preventing disease that considers individual genomic, proteomic, and metabolomic characteristics, as well as lifestyle and environmental influences.

One important area of proteomics is the clinical study of disease (especially cancer) biomarkers and potential drug targets. In the past 10 years, proteomics research has made significant progress [80,81,82,83]. To find specific biomarkers, proteomics researchers usually try to analyze the diversity of the human proteome, which includes multiple proteoforms. Considering the perturbation of protein and proteoform profiles induced by the disease, there is hope in finding disease-specific proteoforms to be used as biomarkers or drug targets. A correlation between exact proteoforms and a given disease phenotype will give us a chance to perform a proteoform-specific assay [84,85]. 

Proteoforms may be more specific markers of body conditions. There are some clinical examples of proteoform usage in clinics. Maybe the best example is a fucosylated form of alpha-fetoprotein (AFP-L3) that is a more reliable biomarker than an unmodified form of AFP for the early diagnosis of hepatocellular carcinoma (HCC). Also, a high level of AFP-L3 has been found in the plasma of patients with various carcinomas [86]. 

The products of protein degradation (the degradome) can also be useful biomarkers, as numerous pathological conditions, including protein aggregation diseases, autoimmunity, and cancer, are accompanied by alterations in protease activity [39]. 

As an important proteomics step in the long-term clinical study of proteoforms, the inventory of proteoforms in normal and cancer cell types and blood plasma is necessary. The Human Plasma Proteome Project (HPPP) was initiated in 2002 “as the means to overcome the major challenges for proteomics studies utilizing blood plasma”. In the last 10 years, significant progress has been made, mainly due to the Consortium for Top-Down Proteomics. The results obtained by the Consortium are being compiled in the Blood Proteoform Atlas (BPA). In the context of liver transplantation, the BPA has been shown to have potential for clinical use based on a proteoform signature that distinguishes normal graft function from acute rejection and other causes of graft dysfunction [59].

The appearance of multiple proteoforms produced by genetic polymorphisms, alternative splicing, PTMs, etc., produces a landscape where some proteoform signatures can be different between the norm and cancer and can be used as specific biomarkers. There is hope that progress in proteomics methods should improve the situation in searching for these biomarkers [26,54,87,88]. Proteomics is generating and analyzing a large volume of data, and these data exactly fit the situation with multiple variations in plasma proteomes during cancer development and progression. Here, high-throughput, quantitative mass spectrometry is the best choice. There is already a good example of the possibility of using it in the clinic [89]. Geyer et al. introduced a rapid and robust “plasma proteome profiling” LC–MS/MS pipeline. Their single-run shotgun proteomics workflow enables quantitative analysis of hundreds of plasma proteins from just 1 μL of plasma [89]. Also, AutoPiMS, a single-ion MS-based multiplexed workflow for top-down tandem MS (MS2), was introduced recently and can be used for the analysis of cancer biopsies in a semi-automated manner. AutoPiMS allowed direct identification of more than 70 proteoforms from human ovarian cancer sections [87]. 

Precision medicine helps health care providers better understand the many things—including environment, lifestyle, and heredity—that play a role in a patient’s health, disease, or condition. According to the Precision Medicine Initiative, precision medicine is “an emerging approach for disease treatment and prevention that takes into account individual variability in genes, environment, and lifestyle for each person”.

As new innovations in proteomics technology are starting to become routine practice in clinics, the proteoform profiles themselves can be used as powerful diagnostic markers in many diseases, including cancer (Figure 7). However, several obstacles remain to be overcome before that happens [90]. The most important is the normalization of proteomics methods for the production of reliable protein and proteoform patterns [91]. Here, artificial intelligence-based methods will provide invaluable assistance. They can especially help gain more insights from the data generated by proteomics techniques. The greatest limitation faced by the proteomics field has been its intricacy. 

## 8. Conclusions and Future Perspectives

Despite the constant efforts to generate a clear definition for a variety of protein forms (proteoforms), some ambiguity exists in this area. It happens partially because of the tight intersection of the proteome and peptidome areas. Sometimes it is difficult to find the point of transition between these kingdoms. But when talking about proteoforms, we need to accept and keep in mind all the nuances that are involved in their formation.

The complexity of different human proteoforms emerging due to PTMs is tremendous, but available information indicates that only a small part of it is being implemented. It seems that there is strict control over the production and maintenance of PTMs. This control can be organism-specific and slightly different for different people. Due to this personal variability of proteoform patterns, in sum, the number of all proteoforms presented in the whole human population could cover all possible proteoform patterns. Future in-person, detailed analyses of proteoform profiles (patterns) could confirm this speculation. Here, the situation is that each protein can have many proteoforms according to sites of modification, but because of slightly different systems of personal regulation, the patterns of PTMs that are realized in each organism are different. Methods are needed that allow targeted identification of proteoforms in complex samples. There is already a good example that describes an approach based on the principles of selected/multiple reaction monitoring (SRM/MRM)—proteoform reaction monitoring (PfRM) [92]. The results provide hope that PfRM has the potential to facilitate accurate quantification of protein biomarkers for diagnostic purposes and improve our understanding of disease etiology at the proteoform level [92].

In conclusion, it seems that there is no immediate, simple answer to the question about the regulation of proteoform variety. The situation will become clearer when more information about proteoform variety in different samples of human origin is obtained. The available proteoform databases gathering this information should play a pivotal role in this process.

## Figures and Tables

**Figure 1 proteomes-12-00015-f001:**
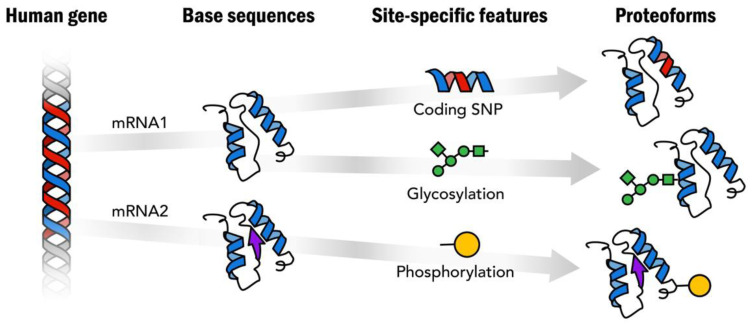
Proteoforms are distinct protein forms arising from a single human gene. Reproduced with permission from [12].

**Figure 2 proteomes-12-00015-f002:**
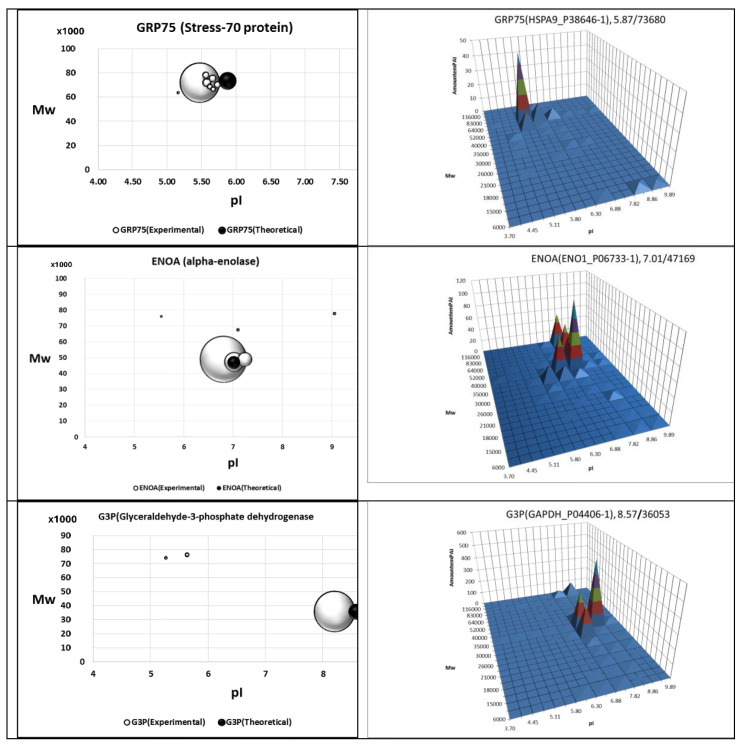
Proteoforms were identified after 2DE separation and following ESI LC–MS/MS analysis. Detection was performed in spots (**left**) or sections (**right**). Proteoform abundance (emPAI) is expressed as a ball size or a peak height. Reproduced with permission from [21].

**Figure 3 proteomes-12-00015-f003:**
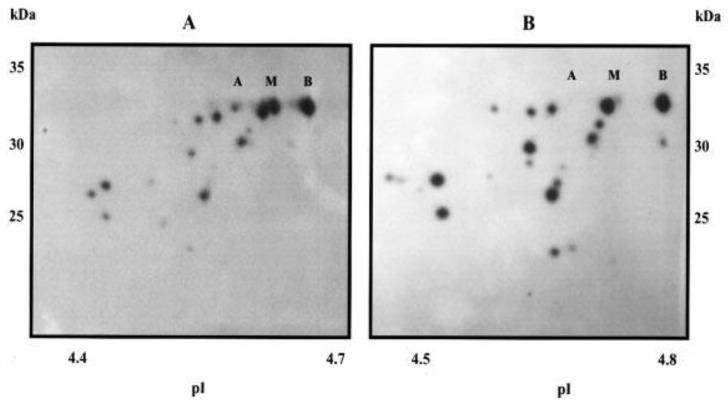
High-sensitivity 2DE Western blots of proteins from the human cells MDA-MB231 (**A**) and the hamster cells CHO (**B**) reveal a proteasomal degradation of proliferating cell nuclear antigen (PCNA). A, M, and B are the full-size proteoforms of PCNA that are usually detected. Reproduced with permission from [34].

**Figure 4 proteomes-12-00015-f004:**
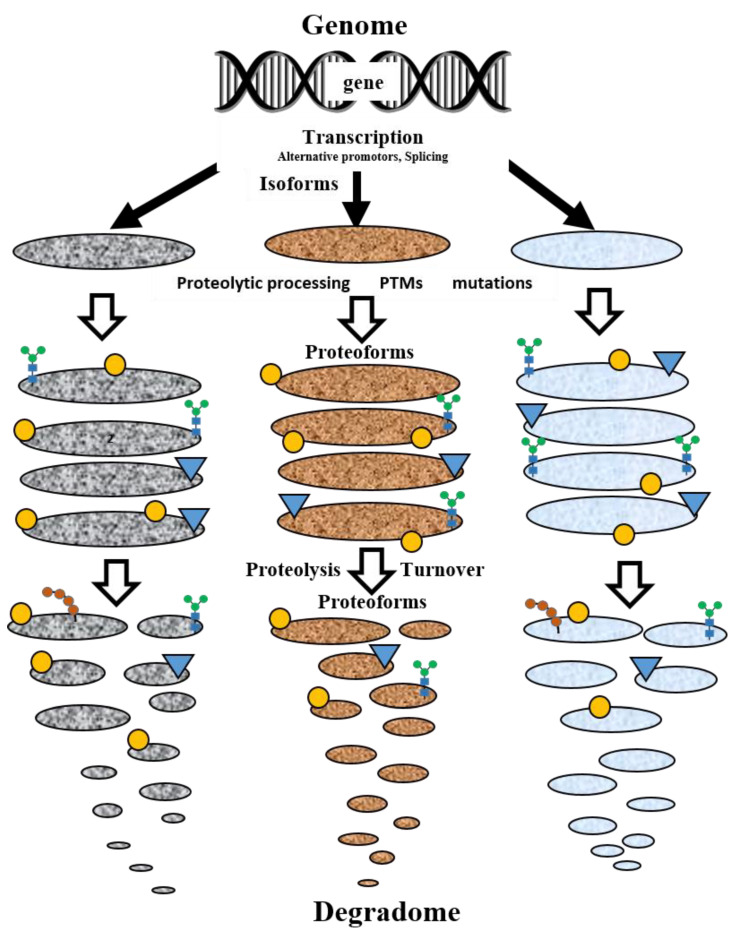
A general scheme of proteoform generation and turnover.

**Figure 6 proteomes-12-00015-f006:**
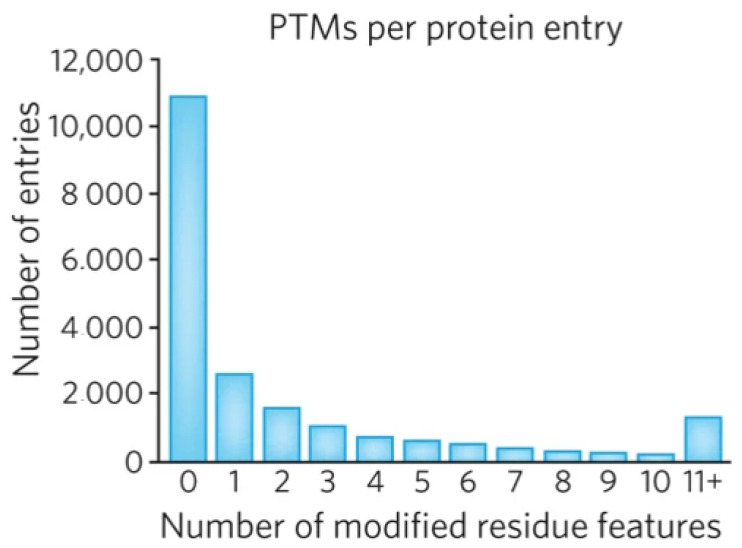
Histogram of PTMs per SwissProt entry. Reproduced with permission from [2].

**Figure 7 proteomes-12-00015-f007:**
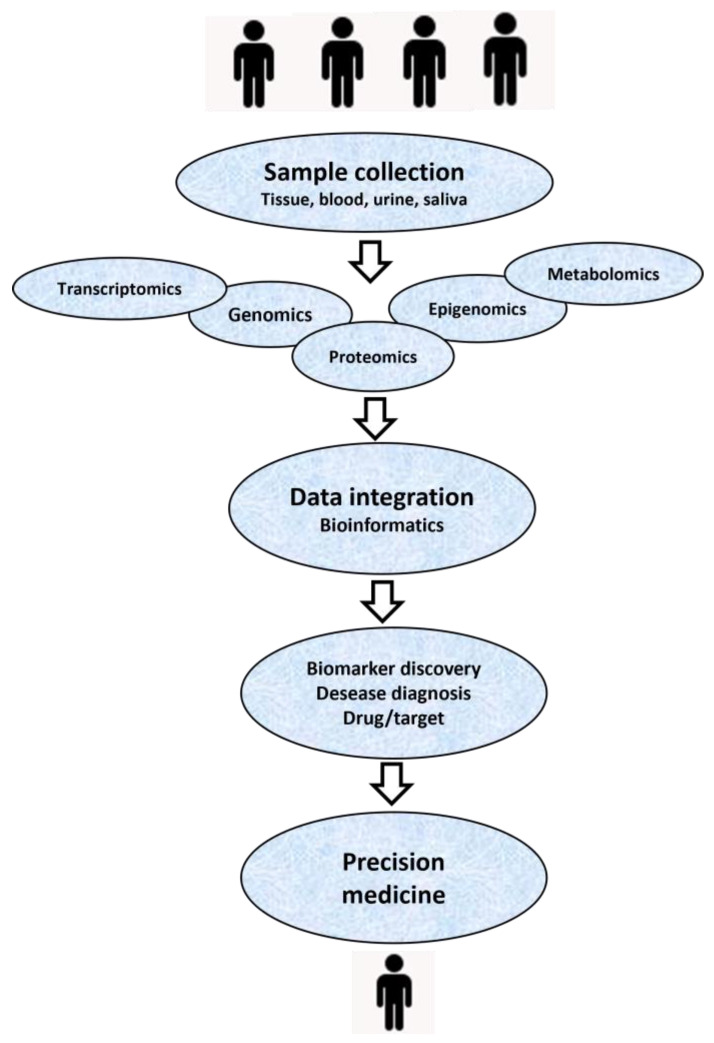
Example workflow for personalized medicine. Once patients have completed all required tests, multi-omics analyses are performed, the results of which are integrated to create individual molecular profiles, including proteoform patterns of specific marker proteins. These profiles are then compared with previously defined biomarker–omics signatures of diseases, which guide treatment selection. Based on this correspondence, the appropriate treatment method is selected.

**Table 1 proteomes-12-00015-t001:** Top proteins having multiple proteoforms according to data from databases “2DE-pattern” and “Proteoform atlas”.

Protein	Gene	Isoform Uniprot #	PTM Sites *	“2DE-Pattern” **	“Proteoform Atlas” ***	Protein Class
H32	*HIST2H3A*	Q71DI3-1	23	21	2979	Chromatin
H4	*HIST1H4A*	P62805-1	38	80	1113	Chromatin
HS90B	*HSP90AB1*	P08238-1	161	82	43	Chaperone
CH60	*HSPD1*	P10809-1	157	48	91	Chaperone
ENOA	*ENO1*	P06733-1	111	78	302	Metabolic
KPYM	*PKM*	P14618-1	132	77	124	Metabolic
G3P	*GAPDH*	P04406-1	120	68	832	Metabolic
PGK1	*PGK1*	P00558-1	97	53	104	Metabolic
LDHA	*LDHA*	P00338-1	72	72	121	Metabolic
EF1A1	*EEF1A1*	P68104-1	105	20	114	Metabolic
HNRPK	*HNRNPK*	P61978-1	132	51	33	RNA metabolism
TBB5	*TUBB*	P07437-1	76	66	63	Cytoskeleton
MYH9	*MYH9*	P35579-1	243	47	115	Cytoskeleton
ACTB	*ACTB*	P60709-1	68	73	1014	Cytoskeleton
VIME	*VIM*	P08670-1	139	65	261	Cytoskeleton
FLNA	*FLNA*	P21333-1	323	57	139	Cytoskeleton
RS27A	*RPS27A*	P62979-1	45	73	65	Ribosomal
1433Z	*YWHAZ*	P63104-1	64	34	125	Scaffold/adaptor

* The data about PTM sites were taken from the database PhosphoSitePlus (https://www.phosphosite.org/) (accessed on 22 December 2023). ** Number of proteoforms according to the database “2DE-pattern” (http://2de-pattern.pnpi.nrcki.ru/) (accessed on 22 December 2023). *** Number of proteoforms deposited in the database “Proteoform Atlas” (http://human-proteoform-atlas.org/proteoforms) (accessed on 22 December 2023).

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
