# Peer review of "Puzzle of Proteoform Variety—Where Is a Key?"

_proteomes, 2024, doi:10.3390/proteomes12020015_

Round 1

Reviewer 1 Report

Comments and Suggestions for Authors

Overall this is an interesting paper that puts together the current problems people encountering for the tremendous number of the protein forms even with the same protein, with the definition evolutions across the time that people want to solve the problems.

1.       One major thing that is not quite clear in this paper is from the title– where the key for the proteoforms is. The article listed the fact that it is difficult to define the protein forms, but there is no clear or well-established answer for how to decipher the complexity. It would be great to better emphasize/illustrate this point.

2.       Another thing is in this paper, one of the proposed idea for better definition is to use 2DE patterns. 2D gel electrophoresis is surely a mature and robust method for protein separation, however, it’s getting outdated today and there are less and less labs in the world are using 2DE. The problem is that even though 2DE pattern may be useful for the protein forms separation,  the lack of usage leads to smaller database for the whole proteome. This is a point that needs to be clarified and further discussed.

3.       In section 3, what is the LC-ELCI-MS/MS?

Reviewer 2 Report

Comments and Suggestions for Authors

In this paper, the author provides a review on previous studies over the complexity of proteoforms in a given cell. The author concludes that there is a control on post translational modifications to generate different proteoforms in each individual and it is personalized (person-specific).

The subject is very interesting and is suitable for the Proteomes. However, I would recommend the author to provide much more literature review and more in depth discussion on this topic before this paper can be considered for publication in Proteomes.

The author emphasizes on the human-specific nature of post translational modification profile for each over protein in a cell to generate a series of proteoforms. The number of proteoforms for a given protein depends on the specific sites of the protein being used for post translational modifications.

The topic is relevant in the field of proteomics. This topic could benefit the development speed in the field of drug and bio-marker discovery and personalized medicine.

The author provides a brief review on the the complexity of different human proteoforms through post translational modifications and highlights that only a small part of the protein is being utilized for this purpose. The author signifies that there is a control over the production and maintenance of post translational modifications are governed by human- and organ-specific proteins.

The following subjects need to be discussed: bioinformatic tools, separations, role of proteomics in drug discovery and disease diagnosis.

The conclusion is consistent with the evidence and arguments presented in the paper. However, more discussion needs to be provided in regards to the research directions in the future in this field.

There must be more recent references regarding this topic.

More figures could be added to illustrate the importance of proteomics in various field of medicine.

Reviewer 3 Report

Comments and Suggestions for Authors

The submitted review “Puzzle of Proteoforms – where is a key?” by Konstantinov BP provides a detailed overview of the complexity of proteomics, covering various aspects of the role of proteoforms in the human proteome. The presented approach ensures a thorough understanding of the challenges creating proteofom databases, as well as the discrepancies between the theoretical and experimentally identified proteoforms.

 The review focuses on several key points – the initiative of the Human Proteome project to create databases of the human proteome, the experimental approaches used to identify PTMs, the discrepancies in terminology, number of proteoforms and the “place” of relevant PTMs in databases created for the proteome vs peptidome vs degradome.

Overall comments:

The authors touch base on the misconception of the human proteome complexity elaborating that recent discoveries have revealed only 19,778 genes, yet surprising level of complexity due to the production of different protein molecules from a single gene, known as proteoforms. Can the authors clarify this statement – since “protein molecules” are reference of proteins, whereas “proteoforms” consider PTMs or different forms of a “protein molecule”?

The authors are encouraged to elaborate more on the complexity at “isoform” level. Protein diversity can be genetically driven and certain isoforms are present in certain populations for selected proteins.

Authors discuss the complexity of quantifying proteoforms and suggest top-down proteomics as favorable, but complementary with bottom-up approaches in PTM detection. Can the authors include the limits of detection and sensitivity of the outlined methods to enable the reader to grasp the complexity of PTM quantification? Also, if they can discuss in more detail the limitations of 2D GE (resolution power) for PTM separation?

Authors present a combinatorial calculation for the theoretical number of PTMs which could be associated with a certain protein. Even though this is a solid mathematical approach, the translation in protein synthesis is not to be taken literally. The authors are encouraged to elaborate on the mechanism of PTM formation in vivo and maybe suggest a more realistic calculation which would account for protein structure, steric effects, and other conformational factors; a model is encouraged.

Authors state that proteoforms may serve as molecular fingerprints, offering insights into individual differences. The authors are encouraged to account the dynamic nature of the human proteome which suggests that proteins have a range of half-lives and can express different PTMs at different time periods, and because of certain medications. Authors are encouraged to consider including longitudinal studies for more comprehensive understanding of PTMs.

Authors make a solid point that proteoforms hold promise as biomarkers for diseases like cancer, with specific proteoform signatures indicative of disease states, and provide references for the subject. Authors are encouraged to include a paragraph where they would discuss protein modifications and PTMs originating from environmental factors and suggest potential ways if/how such modifications would need to be handled in global proteome database.

Authors make a good point about the crosslink between proteome, peptidome and degradome, questioning whether signal peptides or smaller truncations should be included in proteome databases. This is rather a complex question, and further elaboration may be beneficial to clarify the author’s concerns on the matter.

Specific Comments:

Positive Aspects:

·       The article provides a detailed overview of the complexity of proteomics, covering terminology, methodologies, quantification, clinical implications, and future perspectives of the proteome analysis.

·       The article offers valuable insights into the challenges and advancements in proteomics research.

·       The article maintains clarity in its explanations, making it accessible to readers with varying levels of expertise in proteomics.

·       The article effectively integrates references, databases, and scientific organizations, to support its arguments.

Aspects which need to be addressed:

·       Some readers may find certain sections of the article difficult to understand, especially if they lack prior knowledge of the subject. In addition, there is a tendency to repeat certain statements throughout the article.

·       The article contains technical terminology and jargon typical of the field of proteomics, with some questionable sentence structures, which might pose challenges for readers, especially native English readers.

o   Specific examples for direct (word-to-word) translation include:

§  The title: “Puzzle of Proteoforms – where is a key?”

§  “But the terminology is only one side of the situation.”

·       Authors are encouraged to proofread and reword certain sentences.

·       While the article discusses challenges and unresolved issues in proteomics research, it may benefit from a more explicit discussion of the limitations of current methodologies and approaches – mostly discuss sensitivity and abundance of proteins and proteoforms.

In conclusion, while the article effectively covers a wide range of topics in proteomics research, including terminology, methodologies, quantification, and clinical applications, there are areas where it could be further improved to enhance clarity. A more explicit discussion of limitations and potential solutions, along with clearer proposed next steps, would enhance the overall quality and reader engagement of the article.

Comments on the Quality of English Language

Please consider editing the title of the suggested article and make it more sound. 

Please proof read and ensure proper technical support on sentence structure.

Round 2

Reviewer 2 Report

Comments and Suggestions for Authors

The paper is now perfectly suited for publication in Proteomes.

Author Response

Dear Reviewer,

Thank you for cooperation.